# Expression of Checkpoint Molecules in the Tumor Microenvironment of Intrahepatic Cholangiocarcinoma: Implications for Immune Checkpoint Blockade Therapy

**DOI:** 10.3390/cells12060851

**Published:** 2023-03-09

**Authors:** Lara Heij, Jan Bednarsch, Xiuxiang Tan, Mika Rosin, Simone Appinger, Konrad Reichel, Dana Pecina, Michail Doukas, Ronald M. van Dam, Juan Garcia Vallejo, Florian Ulmer, Sven Lang, Tom Luedde, Flavio G. Rocha, Shivan Sivakumar, Ulf Peter Neumann

**Affiliations:** 1Department of Surgery and Transplantation, University Hospital RWTH Aachen, 52074 Aachen, Germany; 2Institute of Pathology, University Hospital RWTH Aachen, 52074 Aachen, Germany; 3Department of Pathology, Erasmus Medical Center Rotterdam, 3000 CB Rotterdam, The Netherlands; 4NUTRIM School of Nutrition and Translational Research in Metabolism, Maastricht University, 6211 ER Maastricht, The Netherlands; 5Department of Surgery, Maastricht University Medical Center (MUMC), 6202 AZ Maastricht, The Netherlands; 6GROW—School for Oncology and Reproduction, Maastricht University, 6211 ER Maastricht, The Netherlands; 7Department of Molecular Cell Biology & Immunology, VU University Medical Center, 1081 HZ Amsterdam, The Netherlands; 8Department of Gastroenterology, Hepatology and Infectious Diseases, University Hospital Duesseldorf, 40225 Duesseldorf, Germany; 9Division of Surgical Oncology, Knight Cancer Institute, Oregon Health and Science University, Portland, OR 97239, USA; 10Department of Oncology, University of Oxford, Oxford OX3 7DQ, UK; 11Kennedy Institute of Rheumatology, University of Oxford, Oxford OX3 7DQ, UK

**Keywords:** cholangiocarcinoma, tumor microenvironment, checkpoint molecules, multiplexed imaging, immunotherapy

## Abstract

*Background*: The tumor microenvironment (TME) in cholangiocarcinoma (CCA) influences the immune environment. Checkpoint blockade is promising, but reliable biomarkers to predict response to treatment are still lacking. *Materials and Methods*: The levels of checkpoint molecules (PD-1, PD-L1, PD-L2, LAG-3, ICOS, TIGIT, TIM-3, CTLA-4), macrophages (CD68), and T cells (CD4 and CD8 cells) were assessed by multiplexed immunofluorescence in 50 intrahepatic cases. Associations between marker expression, immune cells, and region of expression were studied in the annotated regions of tumor, interface, sclerotic tumor, and tumor-free tissue. *Results*: ICCA demonstrated CD4_TIM-3 high densities in the tumor region of interest (ROI) compared to the interface (*p* = 0.014). CD8_PD-L1 and CD8_ICOS densities were elevated in the sclerotic tumor compared to the interface (*p* = 0.011 and *p* = 0.031, respectively). In a multivariate model, high expression of CD8_PD-L2 (*p* = 0.048) and CD4_ICOS_TIGIT (*p* = 0.011) was associated with nodal metastases. *Conclusions:* High densities of PD-L1 were more abundant in the sclerotic tumor region; this is meaningful for the stratification of immunotherapy. Lymph node metastasis correlates with CD4_ICOS_TIGIT co-expression and CD8_PD-L2 expression, indicating the checkpoint expression profile of patients with a poor prognosis. Also, multiple co-expressions occur, and this potentially suggests a role for combination therapy with different immune checkpoint targets than just PD-1 blockade monotherapy.

## 1. Introduction

Cholangiocarcinoma (CCA) is a deadly disease occurring in the liver and is associated with high mortality rates [1]. This cancer type originates from the bile duct epithelium, and the only chance of cure is a complete resection since cancer’s response to therapeutic agents is often limited [2,3,4]. The emerging field of immunotherapy has not yet led to the revolution hoped for, and not all patients seem to respond equally well. For patients not eligible for surgery, treatment options are limited.

Intrahepatic cholangiocarcinoma (iCCA) demonstrates phenotypic differences in immune cell infiltrate, genetics, and stroma [5]. Recent developments in treatment with immune checkpoint inhibitors (ICI) are promising, but not for every patient. Using this mechanism, the host immune cells are re-activated to improve the patient’s immunological response to the tumor. In several cancers, such as melanoma and lung cancers, responses have been impressive, with an increase in life expectancy [6,7]. There is an urgent need for biomarkers to better predict response to immunotherapy [8]. Predictors for an anti-tumor response to ICIs currently are high PD-L1 expression, microsatellite high (MSI-H) cancers, tumor-infiltrating lymphocytes (TILs) at the edge of the tumor, and a high mutational burden (TMB). Unfortunately, even in the presence of one of these markers, not all patients seem to respond to ICIs [9,10].

In the TME, there is a homeostasis favoring either the host immune response or the tumor growth [11]. Here, crosstalk between many cell types takes place, and the immune cell interaction with the cancer is among them [12,13]. In the TME, antigen-presenting cells can express ligands to block the tumor-infiltrating T cells that express co-inhibitory receptors. The pathway behind the immune checkpoint interaction can either be co-stimulatory or co-inhibitory, and this phenomenon is important for an anti-tumor response [14]. It has been shown in several cancers that co-expression of checkpoint molecules on immune cells correlates with survival and is associated with longer progression-free survival after anti-PD-1/PD-L1-based therapies [15,16]. PD-1 and PD-L1 blockade therapy showed a worse progression-free survival in the presence of TIM-3 and CD68 overexpression [15]. Results of a phase 3 trial with the PD-1 blocker Durvalumab in advanced biliary tract cancer patients showed a better overall survival for the group receiving a combination of chemotherapy and immunotherapy [17].

Good biomarkers to predict response to ICIs are still lacking, and even with high expression levels of PD-L1, response rates vary between 37% for combination chemotherapy and immunotherapy [17] and 5.8% for patients treated with Pembrolizumab regardless of their PD-L1 status [18]. Even though it has been known that PD-L1 is not a representative biomarker in advanced biliary tract cancer [19], we aimed to describe differences in PD-L1 expression within one histology slide. This finding can be relevant for the enrollment of patients in future trials when the combined positive score (CPS) is used on a biopsy. In intrahepatic cholangiocarcinoma, the central core sometimes shows a sclerotic, dense stroma with just a few immune cells present, while the interface region is the region where the invasion into the surrounding tissue takes place.

In this study, we have integrated the co-inhibitory checkpoints TIM3, LAG3, CTLA-4, TIGIT, PD-1, PD-L1, PD-L2, and the co-stimulatory checkpoint ICOS. These checkpoint molecules are all in advanced stages of clinical development, and some therapeutic targets are available or in development. The inhibitory checkpoints are regulators of the immune system. PD-L2 is a second ligand for PD-1 and inhibits T-cell activation, just as PD-L1. The PD-1 receptor is expressed on the surface of activated T cells. PD-1 and PD-L1/PD-L2 belong to the family of immune checkpoint proteins that act as co-inhibitory factors that can limit the development of the T cell response. PD-L1 and PD-L2 expressed on the tumor cells bind to PD-1 receptors on the activated T cells, which leads to the inhibition of the cytotoxic T cells. These deactivated T cells remain inhibited in the tumor microenvironment.

Additionally, we analyzed the presence of the immune cells CD4, CD8, and CD68 in primary resected patients with iCCA using multiplexed imaging on whole slide imaging. Our aim was to perform deep immunophenotyping and characterize the immune environment and expression of immune checkpoint differences between ROIs (see Figure 1, Study workflow).

## 2. Materials and Methods

### 2.1. Patient Recruitment

FFPE blocks were collected from the pathology archive, RWTH University Hospital Aachen between 2010 and 2019 from 50 patients diagnosed with iCCA who were fit for surgery with a curative intent; for specific patient characteristics, see Table 1. Of the 50 patients, in 5 individuals ROIs were excluded due to bad quality of the ROI after staining (see Table 2 for the exact numbers of ROIs used in the analysis). The study was conducted in accordance with the requirements of the Institutional Review Board of the RWTH-Aachen University (EK 106/18 and EK 360/19), the Declaration of Helsinki, and good clinical practice guidelines (ICH-GCP).

### 2.2. Sample Collection

Hematoxylin and eosin (H&E) slides were collected, and the slide with the vital tumor and the presence of the interface area was selected by the pathologist (LH). The selected block was used for further processing for our multiplexed imaging workflow using the Tissuefaxs method (TissueGnostics, Vienna, Austria). For tumor staging and grading, the TNM classification was used according to the AJCC/UICC 8th edition.

### 2.3. Whole Slide Multiplexed Immunofluorescence (mIF)

All FFPE samples were subjected to multiplexed immunofluorescence (mIF) in serial 5 μm histological tumor sections obtained from representative FFPE tumor blocks. The FFPE blocks were carefully selected for the presence of the tumor region and, if available, the interface region. The sections were labeled using the Opal 7-Color fIHC Kit (PerkinElmer, Waltham, MA, USA). The antibody fluorophores were grouped into a panel of 5 antibodies. The order of antibody staining was always kept constant on all sections, and sections were initially counterstained with DAPI (Vector Laboratories). The multiplexed immunofluorescence panel consisted of CD4, CD8, CD68, PD-1, PD-L1, PD-L2, ICOS, TIGIT, TIM-3, CTLA-4, and LAG-3 (See Appendix A). All antibodies were diluted with antibody diluent (with background-reducing components, Dako, Germany). Secondary antibodies were applied with the ImmPRESS™ HRP (peroxidase) Polymer Detection Kit (Vector Laboratories, US). TSA reagents were diluted with 1× Plus Amplification Diluent (PerkinElmer/Akoya Biosciences, Waltham, MA USA).

The manual for mIF is described as Edwin R. Parra’s protocol [20]: in short, the first marker was incubated after the FFPE sections were deparaffinized in xylene and rehydrated in graded alcohols. The second marker was applied the following day. And the third marker was applied on the third day. After the five sequential reactions, sections were finally cover-slipped with VECTRASHIELD^®^ HardSet™ Antifade mounting medium.

The slides were then digitally scanned with the TissueFAXS PLUS system (TissueGnostics, Austria). Image analysis was performed in different regions of interest (ROI) in each image (only if present in the slide): tumor, tumor-free, interface, and sclerotic tumor. The size of the ROI varies per slide. Immune cell expression was calculated in percentages throughout the whole project.

Strataquest software was used to analyze the antibody staining and cell counts. The library information was used to associate each fluorochrome component with a mIF marker. All immune cell populations were quantified as positive cells per mm^2^ using cell segmentation, and thresholds were set manually under the supervision of two pathologists (LH/MD). Positive cell counts were categorized based on thresholds; a value above the threshold was considered positive. Checks were performed by the pathologists (LH/MD).

### 2.4. Statistical Analysis

The primary endpoint of this study was to compare immune cell composition, distribution, and co-expression of checkpoints in iCCA. Group comparison was performed on the tumor and interface regions. The tumor ROI was outlined in areas with vital cancer cells, avoiding areas with necrosis. Sclerotic tumor was defined in areas with the presence of dense eosinophilic, sclerotic stroma, usually located in the central area of the tumor only. The interface area was outlined in cases with the presence of the invasive margin of the tumor, usually in the area where the cancer cells invade the surrounding tissue. Tumor-free ROI was outlined in slides with the presence of cancer-free tissue.

The different ROIs were compared with the aim of visualizing distribution differences within one slide. Associations of immune cell subsets with clinical variables were investigated by means of binary logistic regressions. Therefore, immune cell expression data was converted into a dummy variable (high expression vs. low expression) using the median expression as a cut-off for grouping between high and low expression.

Group comparisons were conducted by the Mann–Whitney U test or T-test in the case of continuous variables, while the χ2 test and Fisher’s exact test in accordance with the scale and number count were used in the case of categorical variables. The Wilcoxon matched-pairs test or the paired *t*-test was applied to determine statistically significant differences between values of immune cells within ROIs. The level of significance was set to alpha = 0.05, and *p*-values were calculated using 2-sided testing. All statistical calculations were implemented using Python (v25, IBM, Armonk, NY, USA).

## 3. Results

### 3.1. Patient Characteristics and Clinical Data

The clinical cohort comprised 50 patients with localized iCCA that underwent curative-intent surgery at our hepatobiliary department. Due to our aggressive department standards, a notable number of patients presented with multifocal disease (34%). A subset of patients also underwent neoadjuvant therapy (6%). Major liver resection was the treatment of choice in the majority of the patients (86%), resulting in an R0 resection rate of 92%. Nodal metastases were present in 32% of the overall cohort. Further pathological examination revealed T2 tumors to be the most prevalent (48%) among the patients. The median recurrence-free survival (RFS) of the cohort was 8 months, while the median overall survival (OS) was 32 months. More details regarding the clinical characteristics of the cohort are presented in Table 1.

We first analyzed the immune environment and spatiality; the outlined ROIs were used for comparison: tumor, sclerotic tumor, interface, and tumor-free tissue. The ROIs differed in size; positive cell counting was performed and measured in percentage per mm^2^.

The CD4 and CD8 T cells were more abundant in the tumor area compared to tumor-free tissue (*p* = 0.034 and *p* = 0.0002, respectively). The interface compartment was mainly infiltrated with CD8-positive T cells when compared to tumor and sclerotic tumor ROI (*p* =< 0.001 and *p* = 0.0029, respectively). There was no significant difference in the distribution of CD68 between the different ROIs.

### 3.2. Sclerotic Tumor Demonstrates Higher Expression of CD8 with PD-L1

The comparison of co-expression of checkpoint molecules between the different ROIs illustrated higher levels of CD4 with CTLA-4 in the sclerotic tumor compared to the interface (*p* = 0.02). CD4 with co-expression of TIM-3 was more abundant in the tumor compared to interface (*p* = 0.0137), just as was CD4 with PD-1 and TIM-3 co-expression (*p* = 0.027).

CD8 cells with co-expression were significantly increased in the sclerotic tumor when compared to the tumor. CD8 with ICOS and PD-L1 positive cells were more abundant in the sclerotic tumor (*p* = 0.0309 and *p* = 0.0112 respectively). Moreover, CD8 with PD-1 and PD-L1 and CD8 with PD-1 and PD-L2 co-expression were more abundant in the sclerotic tumor (*p* = 0.0288 and *p* = 0.0411, respectively). See Figure 2 for an overview of the results and Appendix A for all *p*-values between the ROIs in iCCA.

### 3.3. Binary Logistic Regression for Nodal Status with Respect to Immune Cell Subsets

To investigate associations between immune cell subsets and the presence of lymph node metastases, we conducted binary logistic regressions. In iCCA high expression of CD8_PDL1_PDL2 (*p* = 0.041), CD8_PDL2 (*p* = 0.041), CD4_ICOS_TIGIT (*p* = 0.024), CD4_TIGIT (*p* = 0.015), CD8_ICOS_TIGIT (*p* = 0.015), and CD8_ICOS_CTLA (*p* = 0.015) within the tumor and low expression of CD8_CTLA (*p* = 0.026) and CD8_TIM3 (*p* = 0.026) within the normal liver tissue were associated with nodal metastases in iCCA. In a multivariate model, high expression of CD8_PDL2 (*p* = 0.048) and CD4_ICOS_TIGIT (*p* = 0.011) were identified as the two independent predictors of nodal metastases in iCCA. More details are depicted in Table 3.

Various Immune cell subsets were prognostic for the presence of nodal metastases. All immune cell subsets within the tumor, interface and normal liver were investigated but, only Immune cell subsets with a significant association in univariate and multivariate analysis are depicted within the table. Immune cell subsets displaying a *p*-value < 0.1 were transferred into a multivariate model using backward elimination. No multivariate analysis was conducted for perihilar cholangiocarcinoma as only tumor_CD68 was associated with lymph node metastases. For intrahepatic cholangiocarcinoma the following variables were included in the multivariate model: Tumor_CD8_PD-L1_PD-L2, Tumor_CD8_PD-L2, Tumor_CD4, Tumor_CD4_ICOS_TIGIT, Tumor_CD4_TIGIT, Tumor_CD4_TIGIT_CTLA-4, Tumor_CD8_ICOS, Tumor_CD8_ICOS_TIGIT, Tumor_CD8_ICOS_CTLA-4, Liver_CD8_PD-1_PD-L1, Liver_CD68_PD-1_PD-L1, Liver_CD4_CTLA-4, Liver_CD8_CTLA-4, Liver_CD4_TIM-3, Liver_CD8_TIM-3. OR, Odds ratio.

## 4. Discussion

Cholangiocarcinoma is a heterogeneous disease, and therapeutic targets are still evolving. The response and mechanisms to resistance to immunotherapy are likely to be found in the TME, but a lot is still unknown, and good biomarkers to predict which patients will respond are still lacking. Intratumoral genetic heterogeneity plays an important role in the response to treatment [10], and the activation state of immune cell subtypes also plays a role [21,22]. Epigenetic mechanisms regulate how genes are expressed, and by this mechanism, the cancer is able to escape the host immune system [23]. For iCCA, it is described that the CD4 T regulatory cells (Tregs) create a highly immunosuppressive environment [8,24], influencing the anti-tumor response in a negative way. Tumor-infiltrating lymphocytes (TILs), especially CD8 cytotoxic T cells, are described to have a favorable outcome [25,26]. Different immune subtypes for iCCA have been described, and the inflamed subtype with massive T cell infiltration contains 11% of the iCCA cases [27]. It is hypothesized that this subgroup is likely to benefit from immunotherapy.

Here, we investigated the immune cell composition and identified differences in immune checkpoint molecule expression between different ROIs, within the same histology slide. The CD8 cells are most highly expressed in the interface area, the invasive margin for the cancer cells to grow into the surrounding tissue, and here their contribution to supporting the host immune system is needed. There was no significant difference in abundance between the ROIs. An explanation for this could be that macrophages are present in large numbers in normal livers as well. CD68 will also stain the resident Kupffer cells. Macrophages can be recruited to areas with damage, such as inflammation or necrosis, but not all macrophages stained by CD68 are tumor-associated macrophages.

The checkpoint molecules are mostly expressed in the sclerotic tumor and the tumor, suggesting immune checkpoint therapy could have a possible effect here by releasing the blocked CD4 and CD8 cells. The CD4 T cells demonstrated expression of the immune checkpoint molecules TIM-3 and PDL-1, both with an immunosuppressive effect in the tumor and interface ROI. The CD8 T cells expressed PD-1, PD-L1, and PD-L2, all having an immunosuppressive effect as well. Further CD8 T cells with co-expression of ICOS were significantly more present in the sclerotic tumor. The sclerotic tumor lesion is expected to be the older central core of the lesion. ICOS has an immune stimulatory effect, and patients with low levels of ICOS might demonstrate a good response to treatment strategies where the immune anti-tumor response is stimulated [15]. Combined, these results identify an immune suppressive environment in iCCA that may be a target for future immunotherapy with a PD1 blockade.

The co-expressions of the checkpoint molecules on the T cells indicate an immunosuppressive environment in this tumor entity. The checkpoint phenotype demonstrated more PD-1, ICOS, and TIM-3 checkpoint expression on CD8 cells. The fact that the interface is mostly lacking in the expression of the checkpoint molecules is an important finding since this is the area where the cancer cells are most aggressive and invade the surrounding tissue. In order to stop the tumor from growing, the interface is an important target area from a biological point of view. Moreover, the presence of effective immune cells in the surrounding regions of the tumor is an important finding. Further research is needed to investigate if immune cells can be recruited to the interface area when the immune cells are released by ICI therapy.

The zonation in the distribution of the checkpoint molecules could be of importance in deciding to treat patients with a combination of chemotherapy and immunotherapy. Currently, PD-L1 expression is still used as a marker to predict response to immunotherapy. When the CPS score is used for patient stratification and this is done on a biopsy in an advanced setting, potentially, patients get unjustly rejected for the treatment. This is of primary importance, while the use of immunotherapy, either in combination or alone, is justified in an advanced setting when other therapeutic strategies are no longer an option. Our data show that the PD-L1 expression on immune cells is highest in the sclerotic tumor area, usually the older central core of the tumor. It is questionable if this region is sufficiently represented in a core biopsy. See Figure 3 for histology examples of the different outlined regions and the differences in morphology. If a biopsy sample is needed for treatment stratification, the interventional radiologist is aware of this, so the biopsy can be taken in the presence of the central tumor region.

Multivariate analysis identified CD8_PD-L2 and CD4_ICOS_TIGIT to be related to nodal metastases in iCCA. Another group has identified CD8 and Tregs as being associated with poor outcomes and nodal metastases [28]. Our findings still need to be validated in other cohorts, but we consider CD8_PD-L2 and CD4_ICOS_TIGIT to be indicative of a poor prognosis. In iCCA patients with at least one lymph node metastasis following resection, there is a suggested median OS of 7 to 14 months [29,30]. The presence of cancer-positive lymph nodes is a known strong predictor of poor outcomes [31,32].

Another important finding is that expression of multiple checkpoint combinations, not just the overexpression of PD-1, was observed. This finding suggests a different therapeutic approach since this tumor will be PD-1 blockade refractory when treated with PD-1 blockade therapy alone. New generations of immune checkpoints, such as TIGIT and ICOS, are expressed as well. Targeting iCCA could mean that a combination of different immune checkpoint agents is a better approach than just a PD-1 blockade monotherapy. It is of primary importance that these new therapeutic strategies are being tested, potentially in a combination immunotherapy and chemotherapy setting. Further research is needed to validate our results.

## Figures and Tables

**Figure 1 cells-12-00851-f001:**
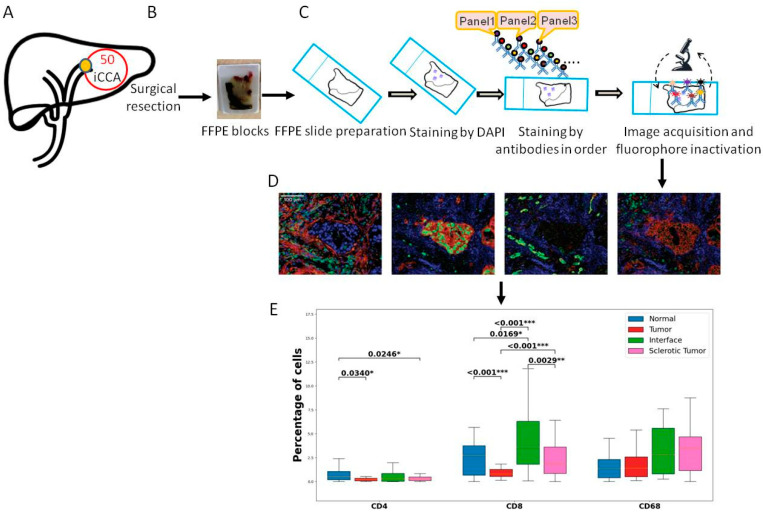
Overview of the study workflow. (**A**) Sample inclusion. We included 50 iCCA cancer samples. (**B**) All patients underwent surgical resection, and one representative FFPE block was collected from the pathology archive. (**C**) Slides were cut and prepared for the multiplex imaging workflow. First staining with DAPI was performed following the antibodies from all panels. Scanning was performed. (**D**) The slides were analyzed using tissuegnostics and software. (**E**) After quantification of the cells and measurement of co-expressions, statistical analysis was performed. Levels of significance: any *p* value less than 0.001 was designated with three (***) asterisks. *p* values between 0.001 and 0.01 are shown with two (**) asterisks and *p* values between 0.01 and 0.05 are shown with one (*) asterisk. iCCA: intrahepatic Cholangiocarcinoma, FFPE: Formalin-fixed paraffin-embedded, DAPI: 4′,6-diamidino-2-fenylindool nuclear staining.

**Figure 2 cells-12-00851-f002:**
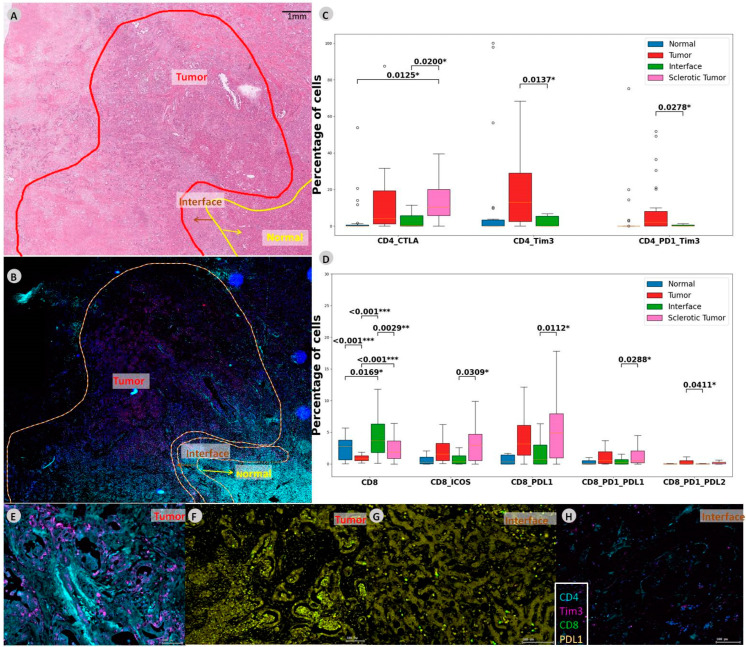
CD4 and CD8 co-expressions in intrahepatic CCA. (**A**) In this slide, different regions are outlined on the HE slide and (**B**) mIF slide. (**C**,**D**) Boxplot demonstrating abundances within the CD4 and CD8 compartments demonstrating the significant abundances of immune cells and checkpoint molecules. (**E**,**F**) Zoomed-in multiplexed image of the outlined tumor ROI. (**G**,**H**) Multiplexed image of the outlined interface ROI. Levels of significance: any *p* value less than 0.001 was designated with three (***) asterisks. *p* values between 0.001 and 0.01 are shown with two (**) asterisks and *P* values between 0.01 and 0.05 are shown with one (*) asterisk. CCA: cholangiocarcinoma, iCCA: intrahepatic Cholangiocarcinoma, ROI: region of interest, CTLA-4: cytotoxic T-lymphocyte-associated protein 4, TIM-3: T-cell immunoglobulin and mucin-domain containing-3, PD-1: programmed cell death protein 1, ICOS: inducible T-cell costimulator, PD-L2: programmed cell death 1 ligand 2.

**Figure 3 cells-12-00851-f003:**
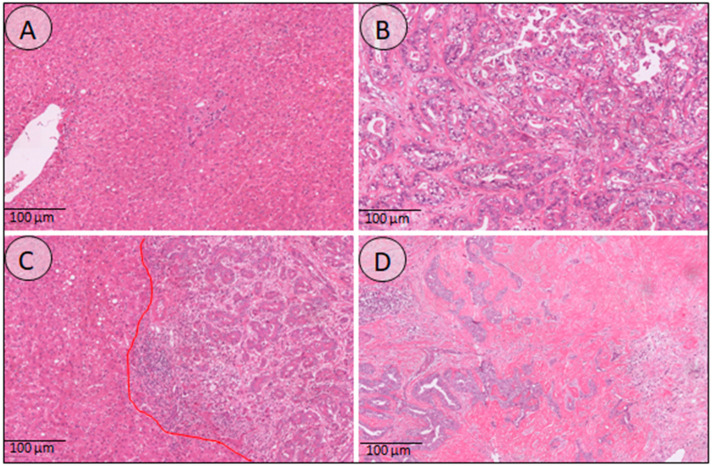
Histology overview of the different annotated regions. (**A**) Normal liver tissue, demonstrating liver parenchyma, portal tracts, and a central vein. (**B**) Tumor region consisting of atypical tumor glands and little desmoplastic stroma in between. (**C**) This image demonstrates the interface region, where the tumor glands grow into the surrounding liver tissue. (**D**) The sclerotic tumor here shows an abundance of dense stroma with just a few tumor glands in between.

**Table 1 cells-12-00851-t001:** Patient characteristics.

	ICCA	
	Total	Percentage
Included patients	50	100%
Gender		
%Male	24	48%
%Female	26	52%
Median age (years)	67	
Multifocal tumor		
%Yes	17	34%
%No	33	66%
Tumor stadium (T)		
%UICC T1	17	34%
%UICC T2	24	48%
%UICC T3	6	12%
%UICC T4	3	6%
Nodal status (N)		
%N0	32	64%
%N1 + N2	16	32%
%not known	2	4%
Tumor grading (G)		
%G1	0	0%
%G2	37	74%
%G3	10	20%
%not known	3	6%
Perineural invasion (Pn)		
%Pn0	10	20%
%Pn1	13	26%
%not known	27	54%
Residual tumor (R)		
%R0	46	92%
%R1	4	8%
Lymphovascular invasion (L)		
%L0	38	76%
%L1	9	18%
%not known	3	6%
Median overall survival (months)	32	

**Table 2 cells-12-00851-t002:** Overview of number of ROIs included in the analysis.

iCCA-ROI	Number
Normal	30
Tumor	45
Interface	30
Scleotic tumor	32

**Table 3 cells-12-00851-t003:** Univariate and multivariate binary logistic regression for nodal status in cholangiocarcinoma with respect to immune cell subsets.

Immune Cell Subset	Perihilar Cholangiocarcinoma		Intrahepatic Cholangiocarcinoma	
	OR (95%CI)	*p*-Value	OR (95%CI)	*p*-Value
	**Univariate analysis**			
Tumor_CD68	4.09 (1.16–14.43)	0.028		
Tumor_CD8_PD-L1_PD-L2			4.72 (1.07–20.89)	0.041
Tumor_CD8_PD-L2			4.72 (1.07–20.89)	0.041
Tumor_CD4_ICOS_TIGIT			5.36 (1.25–23.04)	0.024
Tumor_CD4_TIGIT			6.11 (1.41–26.41)	0.015
Tumor_CD8_ICOS_TIGIT			6.11 (1.41–26.41)	0.015
Tumor_CD8_ICOS_CTLA-4			6.11 (1.41–26.41)	0.015
Liver_CD8_CTLA-4			0.13 (0.20–0.78)	0.026
Liver_CD8_TIM3			0.13 (0.20–0.78)	0.026
	**Multivariate analysis**			
Tumor_CD8_PD-L2			5.24 (1.01–27.18)	0.048
Tumor_CD8_ICOS_TIGIT			8.05 (1.60–40.53)	0.011

## Data Availability

All data generated or analyzed during this study are included in this article. Further enquiries can be directed to the corresponding author.

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
