# Peer review of "Expression of Checkpoint Molecules in the Tumor Microenvironment of Intrahepatic Cholangiocarcinoma: Implications for Immune Checkpoint Blockade Therapy"

_cells, 2023, doi:10.3390/cells12060851_

Round 1
Reviewer 1 Report
The scientific impact of this study is high, the methods used are adequate as well as the representations used. The bibliography used represents the most recent knowledge on the subject. Undoubtedly, this study lays the foundations for further research, as to date there are no good biomarkers for immunotherapy.
Reviewer 2 Report
Here, authors give a brief description of immune cells profiles in intrahepatic cholangiocarcinoma (iCCA) patients by multiplex analysis. Particularly, they observed that there is differences in immune checkpoint molecule expression as following: CD8 cells are elevated in the interface area, and the checkpoint molecules are mostly expressed in the sclerotic tumor and the tumor. At last, the checkpoint molecule, PDL1 is highly expressed on immune cells in the sclerotic tumor area. Overall the study where performed well and give interesting information about immune cells profiling in CCA. However, several concerns should be addressed by the authors:
- Authors should explain which classification they used to perform tumor staging and grading analysis.
- There is a contradiction in the manuscripts about the number of patients in the study. In the results paragraph, authors claim that they analyzed 50 iCCA patients (Table 1), but the immunohistochemistry analysis were performed only on one histology analysis (page 8, line 264, discussion). Authors should clarify this issue. Furthermore, if they analyzed only one histological tissue, how the p values of the manuscript are significant?
- Figure 2, panel C and F: Authors should show all the four mentioned areas that they analyzed: (tumor, sclerotic tumor, interface and tumor-free tissue) to make more clear.
- According to the author, (page 4, line177), they subdivided the CCA-liver biopsy in four parts: Tumor, sclerotic tumor, interface and tumor-free tissue. However, on figure 2, authors showed data either for Tumor, sclerotic tumor, interface or Tumor, sclerotic tumor. They should show all the groups.
- Table 2.3.2 is not showed in the manuscript.
- Authors should show data of CD68 expression. In addition, they should clarify why there is not difference distribution of 68 expression among tumor, sclerotic tumor, interface and tumor-free tissue.
Reviewer 3 Report
The submitted manuscript entitled “ Zonation of PD-L1 and co-stimulatory signals in the tumor microenvironment of intrahepatic cholangiocarcinoma: implications for immune checkpoint blockade therapy” by Lara Heij et al. demonstrated that higher levels of PD-L1 expression were observed in sclerotic tumor region and they also characterized the expression immune checkpoint molecules in clinical intrahepatic cholangiocarcinoma (ICC) tissue samples.
Given the extremely low response rate of ICC to ICB, elucidating the immune signature of infiltrated lymphocytes in clinical ICC would be fundamental to developing novel biomarkers and pertinent therapeutic ICB strategies in ICC, a significant clinical unmet problem.
Overall, this type of observatory study is descriptive, yet the manuscript is well written and obtained data is appropriately discussed with depth. I have several minor suggestions.
Minor points:
1) I recommend amending the title of this paper.
2) To help readers understand, it would be beneficial to provide an elucidation of the rationales and roles of selected checkpoint molecules.
3) It would be advantageous to include images of both tumor and sclerotic tumor regions.
4) The legend of Figure 2 is too long and difficult to decipher. Please make subtitles bold (A, B…) and consolidate it if possible.
5) Please provide an explanation about PD-L2 for readers.
6) Line 80: 5,8% ïƒ 5.8%
Author Response
Please see attachement

Round 2
Reviewer 2 Report
TheAuthors answered all my questions in proper way. However, they performed study only on one tissue liver, and I don’t think that statistically relevant. More repeats by increasing N is necessary.
Author Response
Dear reviewer 2,
Please find our point-to-point reply underneath. We thank reviewer 2 for providing constructive feedback and we were able to improve the manuscript.
- clarify the exact number of samples for each analysis since some
areas were missing. In the material and methods paragraph, they claimed
that they used 50 patients and removed 5 individuals "due to bad
quality of the slide after staining". Hence the data analysis was
performed on 45 patients and not 50 patients.
Thank you for this comment. We included 50 patients and selected one FFPE block with tumor for each patient. We investigated the abundance of immune cells and the co-expression of checkpoints between different ROIs. Regions with tumor, sclerotic tumor, interface, and normal tissue were separately analyzed within one slide for cell counts. Unfortunately, not every slide contained all regions. If the region was present, we annotated it. I total all 50 patients were included and analyzed, but in some ROIs the slide was of poor quality and was excluded for this region. For example, in a few cases the tumor region was floated off or damaged, then we excluded this region due to quality issues. This same slide was still used for the normal, interface annotation (when present).
The analysis has been done on all 50 patients.
iCCA -ROI |
number |
Normal |
30 |
Tumor |
45 |
Interface |
30 |
Scleotic tumor |
32 |
2- In the discussion paragraph (line 289-291), they said:" we
investigated the immune cell composition and identified differences in
immune checkpoint molecule expression between different ROIs, within one
histology slide." If they performed the analysis on 45 patients,
authors should rephrase this sentence and said:" (....) between
different ROIs, within the same histological slide”
We adjusted this sentence in the main manuscript and it is highlighted in yellow.
The authors would like to thank the reviewer.
Round 3
Reviewer 2 Report
authors should add the table of ROIs/number of samples used to answer the reviewer in the material and methods in order to make clear the number of samples used for each ROIs
Author Response
Dear reviewer 2,
De table is now added to the main manuscript under table 2 and highlighted in yellow. Also a reference to table 2 in the main text (methods) is added and highlighted in yellow.
Thank you and best regards, Lara